# Tumor MHC Expression Guides First-Line Immunotherapy Selection in Melanoma

**DOI:** 10.3390/cancers12113374

**Published:** 2020-11-14

**Authors:** Elena Shklovskaya, Jenny H Lee, Su Yin Lim, Ashleigh Stewart, Bernadette Pedersen, Peter Ferguson, Robyn PM Saw, John F Thompson, Brindha Shivalingam, Matteo S Carlino, Richard A Scolyer, Alexander M Menzies, Georgina V Long, Richard F Kefford, Helen Rizos

**Affiliations:** 1Faculty of Medicine, Health and Human Sciences, Macquarie University, Sydney, NSW 2109, Australia; elena.shklovskaya@mq.edu.au (E.S.); jenny.lee@mq.edu.au (J.H.L.); esther.lim@mq.edu.au (S.Y.L.); ashleigh.stewart@mq.edu.au (A.S.); bernadette.pedersen@mq.edu.au (B.P.); richard.kefford@mq.edu.au (R.F.K.); 2Melanoma Institute Australia, The University of Sydney, Sydney, NSW 2065, Australia; Peter.Ferguson@melanoma.org.au (P.F.); Robyn.Saw@melanoma.org.au (R.P.S.); John.Thompson@melanoma.org.au (J.F.T.); brindha@brain-surgeon.com.au (B.S.); matteo.carlino@sydney.edu.au (M.S.C.); Richard.Scolyer@health.nsw.gov.au (R.A.S.); alexander.menzies@sydney.edu.au (A.M.M.); Georgina.Long@melanoma.org.au (G.V.L.); 3Department of Medical Oncology, Chris O’Brien Lifehouse, Sydney, NSW 2050, Australia; 4Faculty of Medicine and Health, The University of Sydney, Sydney, NSW 2006, Australia; 5Department of Tissue Pathology and Diagnostic Oncology, Royal Prince Alfred Hospital and New South Wales Health Pathology, Sydney, NSW 2050, Australia; 6Department of Melanoma and Surgical Oncology, Royal Prince Alfred Hospital, Sydney, NSW 2050, Australia; 7Department of Neurosurgery, Chris O’Brien Lifehouse, Sydney, NSW 2050, Australia; 8Department of Neurosurgery, Royal Prince Alfred Hospital, Sydney, NSW 2050, Australia; 9Department of Medical Oncology, Crown Princess Mary Cancer Centre, Westmead Hospital, Westmead, NSW 2145, Australia; 10Department of Medical Oncology, Blacktown Cancer and Haematology Centre, Blacktown Hospital, Sydney, NSW 2148, Australia; 11Department of Medical Oncology, Northern Sydney Cancer Centre, Royal North Shore Hospital, Sydney, NSW 2065, Australia; 12Department of Medical Oncology, Mater Hospital, Sydney, NSW 2060, Australia

**Keywords:** immune checkpoint blockade, anti-PD-1, combination immunotherapy, major histocompatibility (MHC) class I, metastatic melanoma

## Abstract

**Simple Summary:**

Immunotherapy leads to durable responses in a proportion of patients with advanced melanoma. Combination immunotherapy is more efficacious than single-agent immunotherapy, yet it is associated with significant toxicity. Currently there are no robust biomarkers to guide first-line immunotherapy selection. We have developed a flow cytometry-based score, to quantify the expression of antigen-presenting molecules MHC-I and MHC-II on melanoma cells, that incorporates both the fraction of tumor cells expressing MHC molecules and the level of expression. We demonstrate that the evaluation of tumor cell surface MHC-I expression aids in treatment selection, with combination immunotherapy providing clinical benefit over single-agent immunotherapy in MHC-I low melanoma with poor immune cell infiltration.

**Abstract:**

Immunotherapy targeting T-cell inhibitory receptors, namely programmed cell death-1 (PD-1) and/or cytotoxic T-lymphocyte associated protein-4 (CTLA-4), leads to durable responses in a proportion of patients with advanced metastatic melanoma. Combination immunotherapy results in higher rates of response compared to anti-PD-1 monotherapy, at the expense of higher toxicity. Currently, there are no robust molecular biomarkers for the selection of first-line immunotherapy. We used flow cytometry to profile pretreatment tumor biopsies from 36 melanoma patients treated with anti-PD-1 or combination (anti-PD-1 plus anti-CTLA-4) immunotherapy. A novel quantitative score was developed to determine the tumor cell expression of antigen-presenting MHC class I (MHC-I) molecules, and to correlate expression data with treatment response. Melanoma MHC-I expression was intact in all tumors derived from patients who demonstrated durable response to anti-PD-1 monotherapy. In contrast, melanoma MHC-I expression was low in 67% of tumors derived from patients with durable response to combination immunotherapy. Compared to MHC-I high tumors, MHC-I low tumors displayed reduced T-cell infiltration and a myeloid cell-enriched microenvironment. Our data emphasize the importance of robust MHC-I expression for anti-PD-1 monotherapy response and provide a rationale for the selection of combination immunotherapy as the first-line treatment in MHC-I low melanoma.

## 1. Introduction

Immunotherapy has transformed the clinical management of advanced melanoma, resulting in durable responses and improved patient survival. Immune checkpoint inhibitors directed against cytotoxic T-lymphocyte-associated protein 4 (CTLA-4) and programmed cell death protein-1 (PD-1) on T-cells, facilitate immune recognition of tumor neo-antigens and immune-mediated tumor control [1,2,3,4]. Combination (anti-PD-1 plus anti-CTLA-4) immunotherapy results in higher objective response rates (58% as compared to 45% for anti-PD-1 and 19% for anti-CTLA-4 alone) and superior progression-free and overall survival [5]. However, severe (grade 3 or 4) immune-related toxicities are also 1.6 times more frequent with combination immunotherapy compared to anti-PD-1 monotherapy, often resulting in treatment discontinuation [2,3]. Patients with asymptomatic brain metastases, BRAF^V600^-mutant melanoma or elevated serum lactate dehydrogenase (LDH) derive an incremental benefit from combination immunotherapy when compared to single agent anti-PD-1 [5,6], however routine upfront combination immunotherapy for stage IV melanoma patients is not cost-effective [7]. Thus, there is an unmet clinical need to accurately select melanoma patients who require combination rather than single agent immunotherapy based on objective criteria, in addition to the above-mentioned clinicopathologic features.

We recently reported that transcriptional downregulation of the major histocompatibility complex class I (MHC-I) molecules on melanoma cells was associated with melanoma de-differentiation and resistance to anti-PD-1 monotherapy [8]. In this study, we extended our analyses to include pretreatment tumor samples derived from patients who received combination (anti-PD-1 plus anti-CTLA-4) immunotherapy. We employed novel flow cytometry tools to evaluate MHC-I, MHC-II and PD-L1 expression on tumor cells, and correlated expression data with tumor immune contexture. We found that downregulation or loss of MHC-I molecules on melanoma cells is associated with low MHC-II and PD-L1 expression and poor T-cell infiltration into the tumor. Significantly, although these characteristics were associated with lack of response to anti-PD-1 monotherapy, they did not preclude response to combination immunotherapy. Thus, melanoma patients with low MHC-I expression on tumor cells may preferentially benefit from first-line combination immunotherapy.

## 2. Results

### 2.1. Disease Characteristics and Response Assessment

The study included pre-treatment melanoma biopsies from 36 patients subsequently treated with anti-PD-1 (17/36, 47%) or combination (anti-PD-1 plus anti-CTLA-4) immunotherapy (19/36, 53%) (Table 1 and Figure 1a). In this cohort, patients receiving combination immunotherapy were younger (median age 56 versus 70) and 47% had BRAF^V600^-mutant melanoma compared to 12% for anti-PD1 monotherapy. Prior treatment was administered in 3/17 (18%) patients who received anti-PD-1 alone (Ipilimumab in two patients and combination BRAF/MEK inhibitor in one patient), whereas no patients who received combination immunotherapy had prior systemic therapy. Time from biopsy to the start of treatment was 0–328 days (median, 85 days for anti-PD-1 and 98 days for combination immunotherapy).

### 2.2. Response to Combination Immunotherapy in MHC-I Low Melanoma

To examine criteria that may govern differential responses to single agent versus combination immunotherapy, we employed flow cytometry to evaluate expression of the antigen-presenting molecule MHC-I on melanoma cells (Appendix A, Figure 1b). The expression of MHC-I on tumor cells is critical for CD8 T-cell receptor engagement and the resultant tumor killing [9], and melanoma loss of MHC-I has been associated with tumor progression [8,9].

To enable quantitative comparisons across multiple experiments, we applied a flow cytometric quantitative score that reflects both the fraction of tumor cells expressing the marker, and the level of marker expression (Figure 1b). The validity of this assay was confirmed with the identification of MHC-I loss (MHC-I score <0.001, Figure 1c) in independent melanoma biopsies taken from two patients who progressed while on anti-PD-1 treatment. Loss of MHC-I expression in both tumors was associated with loss of the MHC-I invariant chain B2M in the corresponding biopsy-derived melanoma cell lines (Figure 1d). Robust MHC-I expression (MHC-I score ≥1.0, where 1.0 is equivalent to MHC-I expression on 100% of melanoma cells at similar expression levels to peripheral blood mononuclear cells derived from a single healthy donor, used as a comparator across all analyses) was found in all patients who responded (CR/PR) to anti-PD-1 monotherapy, but in only five out of 15 (33%) patients responding to combination immunotherapy, while expression in non-responders was variable (Figure 1e). MHC-I expression correlated with tumor surface expression of MHC class II (MHC-II) (Figure 1f, Appendix A) and the PD-1 ligand PD-L1 (Figure 1f, Appendix A), with considerable yet not complete overlap between MHC-I, MHC-II and PD-L1 low/negative tumors (Figure 1g). All tumors in anti-PD-1 responder group were positive for MHC-I and the majority (90%) also expressed MHC-II and PD-L1, whereas 60% of tumors in combination immunotherapy responder group showed low/absent MHC-I, MHC-II and PD-L1 expression (Appendix A). 

To gain insight into the immune environment of MHC-I low tumors, we examined baseline T-cell infiltration in two patients (22 and 32), both with advanced BRAF^V600^-mutant melanoma, low melanoma MHC-I expression (MHC-I score <0.1 in both pre-treatment biopsies, Figure 2a), but distinct responses to immunotherapy. While patient 22 showed complete response following a single dose of combination Ipilimumab and Nivolumab, patient 32 had disease progression on first restaging at 3 months, after 4 cycles of Pembrolizumab. Both patients developed immune-related adverse events in response to immunotherapy, dermatitis, hepatitis, and thyroiditis (patient 22) or vitiligo (patient 32). The pre-treatment biopsies displayed similar patterns of immune cell infiltration, including effector (CD45RO^+^) CD8 T-cells co-expressing the EOMES transcription factor, multiple inhibitory receptors (PD-1, CTLA-4, TIM3, CD244, CD39) and markers of proliferation (Ki-67) and activation (ICOS, 4-1BB) (Figure 2b–c), phenotypes previously associated with response to anti-PD-1 [10,11,12,13]. The effector (CD45RO^+^) CD4 T-cells also exhibited similar phenotypes in both biopsies (Figure 2d). Thus, whereas low MHC-I expression likely contributed to tumor escape from Pembrolizumab-induced T-cell activation in patient 32, it did not preclude response to combination immunotherapy in patient 22.

Analysis of a previously published melanoma proteomics dataset [14] also confirmed that expression of the MHC-I invariant chain B2M was significantly higher in pre-treatment biopsies derived from 39 melanoma patients responding to anti-PD-1 monotherapy compared to 33 non-responding patients (Appendix A). Taken together, these data confirm the importance of intact MHC-I expression for response to anti-PD-1 monotherapy and emphasize that combination immunotherapy could provide clinical benefit in MHC-low melanoma.

### 2.3. Clinicopathologic Features and Immune Contexture of MHC-I Low Tumors

We next examined tumor microenvironment and clinicopathologic features associated with MHC-I downregulation. The cohort was dichotomized into melanoma MHC-I low (MHC-I score <1.0, *n* = 14) and MHC-I normal/high (MHC-I score ≥1.0, *n* = 22). Clinical characteristic most strongly associated with MHC-I downregulation was high serum LDH (Table 2 and Figure 1a). LDH ≥1.5 upper limit of normal was found in 5/14 (36%) patients with MHC-I low tumors compared with only 1/22 (5%) for MHC-I normal/high tumors (*p* = 0.02, Fisher’s exact test). Tumor immune (CD45^+^) infiltration was positively correlated with MHC-I expression (Figure 3a). Among the eleven major immune subsets analyzed (Appendix A), total T-cells expressing the αβ T cell receptor (TCR αβ) and CD8 T-cells were higher in MHC-I normal/high compared to MHC-I low tumors (Figure 3b, Appendix A), while myeloid cells (macrophages/monocytes and myeloid dendritic cells) were enriched in MHC-I low tumors, reflecting a shift from the T-cell-enriched to a myeloid cell-enriched immune environment. Thus, low tumor MHC-I expression was associated with myeloid predominance and reduced tumor T-cell content. 

Surprisingly, although PD-L1 and MHC-I expression on tumor cells was strongly correlated, PD-L1 was highly expressed by tumor-infiltrating lymphocytes and myeloid cells in both MHC-I high and MHC-I low tumors (Figure 3c). Finally, we found that that regulatory T-cells were relatively enriched in combination immunotherapy responders compared to anti-PD-1 monotherapy responders (Figure 3d), and this difference could not be attributed to the variation in tumor MHC expression, nor did it correlate with the relative myeloid fraction. The ratio also trended higher in both non-responder groups, but the difference was not significant. These data suggest that combination immunotherapy may provide therapeutic benefit in MHC-I low tumors by targeting tumor-infiltrating regulatory T-cells.

## 3. Discussion

In this study, we employed quantitative multiparameter flow cytometry to assess expression of antigen-presenting molecules MHC-I and MHC-II on melanoma cells in pre-treatment biopsies from patients subsequently receiving anti-PD-1 or combination (anti-PD-1 plus anti-CTLA-4) immunotherapy. Using MHC scores that reflect both the fraction of tumor cells expressing the marker and the level of expression, we found that intact expression of the MHC-I and MHC-II molecules on tumor cells was strongly associated with response to anti-PD-1 monotherapy, but was not required for response to combination immunotherapy. In particular, two thirds of patients who responded (PR/CR) to combination immunotherapy displayed low MHC-I and MHC-II expression on melanoma cells, a feature that we show is associated with poor baseline prognostic characteristics, including elevated LDH. This is in line with standard clinical practice that selects patients with poor prognostic features, including elevated LDH, for the more toxic combination immunotherapy over anti-PD-1 monotherapy. Although mechanisms linking elevated LDH to poor prognosis are incompletely understood, LDH-associated lactic acid may impede interferon-gamma production by tumor-infiltrating T-cells and NK cells thus interfering with immune-mediated tumor control [15].

PD-1 blockade relies on “re-invigoration” of PD-1^high^ effector CD8 T-cells [10,11] specific for melanoma neo-antigens presented on MHC-I. Accordingly, tumor MHC-I downregulation or loss has emerged as a mechanism of acquired resistance to anti-PD-1 monotherapy [9]. We recently confirmed MHC-I downregulation as a common mechanism of resistance to anti-PD-1 monotherapy [8]. Our current findings are consistent with the importance of MHC-I expression on tumor cells for response to PD-1 inhibitor monotherapy. We now extend these findings by demonstrating that MHC-I expression is not required for response to combined PD-1 and CTLA-4 blockade. 

There is some controversy regarding MHC-I expression on melanoma cells as a potential predictor of anti-PD-1 monotherapy response. Expression of MHC-I signature proteins was a strong predictor of response to anti-PD-1 by proteomics analysis [14] but not by gene expression analysis [16,17], although MHC-I gene expression did trend higher in anti-PD-1 responders compared to non-responders in both studies and amplifications in MHC-I related genes were observed exclusively in responders to anti-PD-1 monotherapy [16]. Our current findings confirm the importance of intact MHC-I expression on tumor cells for response to anti-PD-1 monotherapy and indicate that combination immunotherapy could overcome MHC-I downregulation or loss on tumor cells. These data are consistent with Rodig et al. (2018) who reported that MHC-I and MHC-II expression on pretreatment tumors by IHC was not predictive of response to combination immunotherapy in the CheckMate 069 study [18]. 

Rodig et al. also found that melanoma MHC-II but not MHC-I expression by immunohistochemistry (IHC) was predictive of week-13 response to anti-PD-1 monotherapy in the CheckMate 064 trial [18]. We now confirm that expression of MHC-I and MHC-II proteins on tumor cell surface is highly correlated, as was previously observed at both the transcript and protein levels [19]. While it is difficult to directly compare flow cytometric- and immunohistochemistry-based estimates of MHC expression, we found that IHC-defined thresholds associated with immunotherapy response, 30% for MHC-I [18] and 1% [18] or 5% for MHC-II [20], identified only 43% of MHC-I low tumors in our study and 50% or 83% of MHC-II low tumors, respectively. These results suggest that the differences in the staining and scoring of MHC molecules [20], may account for the discrepancy in the conclusions regarding in particular MHC-I expression as a determinant of anti-PD-1 monotherapy response. We also demonstrate a tight correlation between MHC-I and PD-L1 proteins on tumor cells, potentially providing an insight into the validity of PD-L1 expression as a predictive tool. PD-L1 expression on melanoma cells is often regarded as a surrogate of interferon-γ driven immune inflammation and by extension, the ongoing anti-tumor T-cell response [21]. While tumor positivity for PD-L1 on immunohistochemistry alone was not predictive of patient outcomes [5,22], there was an association between higher melanoma PD-L1 expression scores and improved survival of patients receiving anti-PD-1 monotherapy [23]. However, PD-L1 regulation in melanoma cells is exceedingly complex and PD-L1 expression may be driven by tumor-cell intrinsic mechanisms in a proportion of patients who exhibit or develop resistance to anti-PD-1 monotherapy [8,24]. To this end, we demonstrate a discrepancy in PD-L1 expression by tumor cells and immune cells within individual biopsies, indicative of differential regulation in melanoma and immune cells in response to the same microenvironmental cues. 

Finally, it is not clear how combination immunotherapy provides clinical benefit in MHC-low, myeloid cell-enriched tumors. Tumor MHC-II expression was proposed to stimulate cytotoxic CD4 T-cell responses in a subset of patients [18]. Although the development of such cells has been documented in animal models and linked to regulatory T-cell depletion by anti-CTLA-4 [25], there is no indication that this mechanism also operates in humans [26]. We found that regulatory T-cell/CD8 T-cell ratio was significantly elevated in combination immunotherapy responders, suggesting that combination immunotherapy provides therapeutic benefit via regulatory T-cell targeting. Anti-CTLA-4-induced blockade of regulatory T-cell function may contribute to the recruitment of novel adaptive immune specificities, as evidenced by the frequent development of autoimmune complications on combination immunotherapy [2], and some of these T-cell clones may eliminate tumor indirectly, such as by targeting the microenvironment [27]. Alternatively, combination immunotherapy may reduce metabolic fitness of immunosuppressive myeloid populations in the tumor microenvironment by targeting regulatory T-cells [28]. Irrespective of the underlying mechanism, our results emphasize the importance of intact tumor MHC-I expression for anti-PD-1 monotherapy response and indicate that combination immunotherapy may provide clinical benefit irrespective of tumor MHC expression and immune contexture.

## 4. Materials and Methods 

### 4.1. Study Approval and Patient Details

Patients with unresectable stage III or IV melanoma treated with immune checkpoint inhibitors at Melanoma Institute Australia and affiliated hospitals between May 2015 and August 2018, with available enzymatically dissociated tumor tissue banked at the time of biopsy, were included in this study. Written consent was obtained from all patients (Human Research ethics committee protocols from Royal Prince Alfred Hospital; Protocol X15-0454 & HREC/11/RPAH/444). Tumor response was assessed using the immune-related response criteria (irRC) [29]. Patients were treated with anti-PD-1 (Pembrolizumab or Nivolumab) monotherapy at currently approved doses, alone or in combination with anti-CTLA-4 (Ipilimumab). All biopsies were assessed by an experienced pathologist prior to tissue banking, and the presence of melanoma cells was confirmed by histopathology. 

### 4.2. Tissue Processing and Flow Cytometry

Tumor biopsies were enzymatically processed as previously described [8]. Briefly, cryopreserved tumor dissociates were thawed and immediately stained with a mixture of fluorescently labeled monoclonal antibodies (detailed in Appendix A) and Fc block to prevent non-specific staining due to Fc receptor binding. Cells were fixed, permeabilized and stained with fluorescently labeled antibodies against intracellular proteins plus Fc block in permeabilization buffer. Live Dead near-infrared (NIR) fixable dye (Invitrogen, Thermo Fisher Scientific, Waltham, MA, USA) was used to exclude non-viable events. Samples were acquired on BD LSR-II Fortessa X20 flow cytometer (BD Biosciences, Franklin Lakes, NJ, USA) and analyzed using FlowJo software v10.5 (TreeStar, Ashland, OR, USA). All events were collected, and samples containing less than 200 viable melanoma cells were excluded from the study. 

Relative MHC-I expression (rMHC-I) was calculated as the geometric mean fluorescence intensity (MFI) of MHC-I on melanoma cells divided by MHC-I MFI on donor PBMC. The MHC-I expression score was then defined as rMHC-I × % marker positive-melanoma/100. MHC-II expression was calculated as for MHC-I. Normal or high MHC-I expression was defined by an MHC-I score ≥1.0, where 1.0 is equivalent to MHC-I expression on 100% of melanoma cells at similar expression levels to control PBMCs. MHC-II positivity was defined by an MHC-II score ≥0.1, where 0.1 is equivalent to MHC-II expression on 2% of melanoma cells at similar expression levels to control PBMCs. Expression of PD-L1 and PD-L2 was calculated as a ratio of PD-L1 or PD-L2 MFI, divided over MFI of the fluorescence minus one control (FMO, background fluorescence in the respective channel after omission of PD-L1 and PD-L2 antibodies). PD-L1 expression score was calculated as rPD-L1 × % marker-positive melanoma/100. 

### 4.3. Cell Culture and Immunoblotting

Short-term melanoma cell lines were maintained in Dulbecco’s Modified Eagle media supplemented with 10% heat inactivated fetal bovine serum (FBS; Sigma-Aldrich, St Louis, MO, USA), 4 mM glutamine (Gibco, Thermo Fisher Scientific, Waltham, MA, USA), and 20 mM HEPES (Gibco), at 37 °C in 5% CO_2_. Cell authentication and profiling was confirmed using the StemElite ID system from Promega. All cells tested negative for mycoplasma (MycoAlert Mycoplasma Detection Kit, Lonza, Basel, Switzerland). For immunoblotting, total cellular proteins were extracted at 4 °C using RIPA lysis buffer containing protease inhibitors and phosphatase inhibitors (Roche, Basel, Switzerland). Proteins (40 µg) were resolved on 12% SDS-polyacrylamide gels and transferred to Immobilon-FL membranes (Millipore Sigma, Burlington, MA, USA). Western blots were probed with antibodies against B2M (1:1000; D8P1H; Cell Signaling Technology, Danvers, MA, USA; Cat No. 12851) and β-Actin (1:6000; AC-74; Sigma-Aldrich; Cat No. A5316).

### 4.4. Statistical Analyses 

Patient demographics and clinicopathologic features including Eastern Cooperative Oncology Group (ECOG) performance status, LDH levels at baseline, mutation status and American Joint Committee on Cancer (AJCC) 8th Edition M stage [30] were collected according to treatment received (single agent PD-1 versus combination immunotherapy) and MHC-I expression (MHC-I high/normal versus low). Fisher’s exact test was used to compare the two groups. Progression free survival (PFS) and overall survival (OS) were described via the Kaplan–Meier method from the start of therapy to the date of last follow-up, disease progression, or date of death. Statistical analyses were carried out using GraphPad Prism software v.8.1.1. Figure legends specify the statistical analysis used and define error bars.

## 5. Conclusions

In this study, we have developed a novel flow cytometry-based scoring system that accurately measures the cell surface expression of antigen-presenting molecules, namely MHC-I and MHC-II, on melanoma cells in patient-derived tumor biopsies. Using MHC scores that reflect both the fraction of tumor cells expressing the marker and the level of expression, we found that the intact expression of MHC-I molecules on tumor cells was strongly associated with a response to anti-PD-1 monotherapy, but was not required for a response to combination immunotherapy. Our data indicate that the flow-based assessment of MHC-I expression in pretreatment melanoma biopsies may aid in first-line immunotherapy selection, by identifying patients with MHC-I-low melanoma that may preferentially benefit from the more toxic combination (anti-PD-1 plus anti-CTLA-4) immunotherapy.

Mechanistically, we have found that pretreatment tumors derived from combination immunotherapy responders are relatively enriched in suppressive regulatory T-cells, suggesting that combination immunotherapy provides therapeutic benefit via direct or indirect regulatory T-cell targeting. 

## Figures and Tables

**Figure 1 cancers-12-03374-f001:**
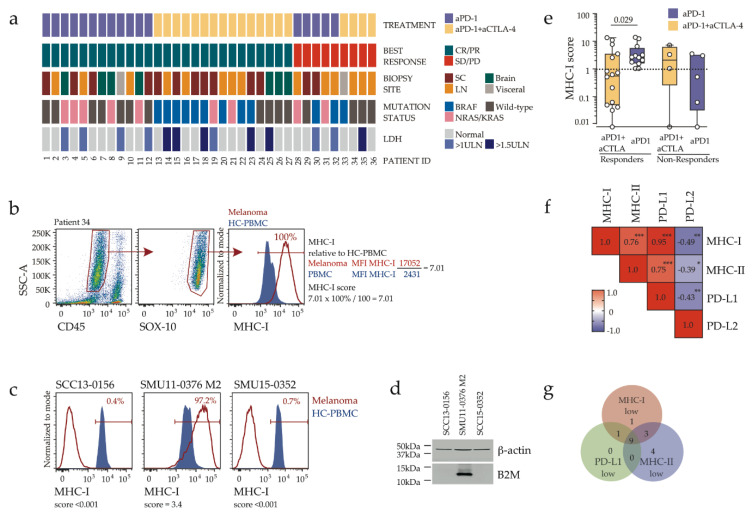
Tumor expression of MHC-I is associated with response to PD-1 inhibitor monotherapy, but not combination immunotherapy. (**a**) Details of the 36 melanoma pre-treatment biopsies analyzed in this study, including 17 from patients subsequently treated with anti-PD-1 (aPD-1) and 19 from patients treated with combination (aPD-1+aCTLA-4) immunotherapy. Patient responses as defined by RECIST criteria, biopsy site, driver mutations and LDH status are shown. CR, complete response; PR, partial response; SD, stable disease; PD, progressive disease; LN, lymph node; SC, subcutaneous, Visceral, non-brain visceral, ULN, upper limit of normal. (**b**) Representative flow cytometry analysis showing melanoma cell gating and calculation of the MHC-I expression score. Histograms show expression of MHC-I on melanoma cells (red) and healthy control peripheral blood mononuclear cells (HC-PBMC; blue shaded histogram). (**c**) Lack of tumor MHC-I expression in two B2M-deficient melanoma biopsies, with MHC-I score indicated. An MHC-I positive melanoma tumour was included for comparison. (**d**) B2M protein expression in short-term melanoma lines derived from the melanoma biopsies shown in (**c**); kDa, kilodalton. (**e**) Melanoma MHC-I expression score, stratified by therapy and patient’s response. Boxes show the interquartile range and the median, with Mann-Whitney’s *p* value indicated. Dotted line indicates positivity threshold (=1.0). Values below 0.01 are shown as zero. (**f**) Correlation matrix of melanoma cell expression of MHC-I, MHC-II, PD-L1 and PD-L2. Spearman’s rank correlation values are shown within the matrix, with asterisks indicating the *p* values (* *p* < 0.05; ** *p* < 0.01; *** *p* < 0.001). (**g**) Venn diagram showing the distribution of tumors with low/negative MHC-I (score <1.0, *n* = 14), low/negative MHC-II (score <0.1, *n* = 16) and low/negative PD-L1 expression (score <1.0, *n* = 10).

**Figure 2 cancers-12-03374-f002:**
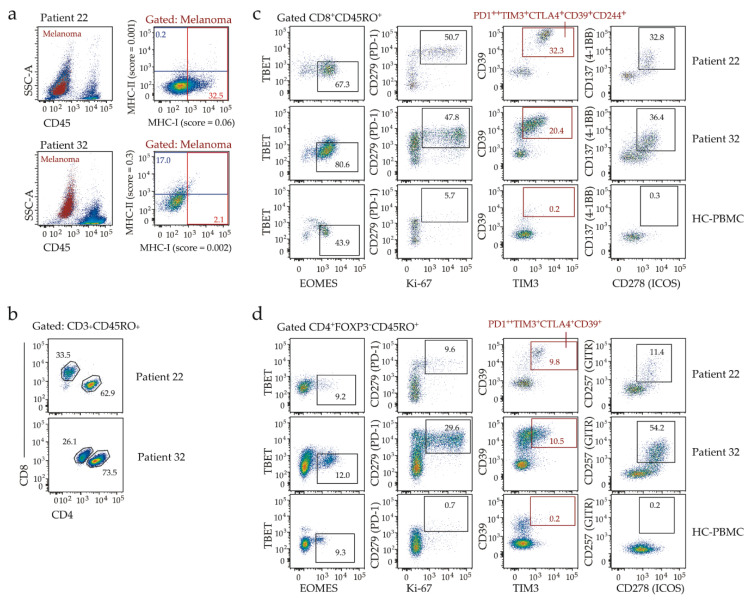
Flow cytometric analysis of pretreatment biopsies from patient 22 and patient 32. (**a**) Expression of MHC-I and MHC-II on melanoma cells. Left, melanoma cells gated as per Appendix A (red) overlaid onto total live events (pseudocolor plot). Right, melanoma MHC-I and MHC-II expression, with percent positivity and expression scores indicated (red boxes, MHC-I; blue boxes, MHC-II). (**b**) Flow cytometric analysis of T-cell subsets, gated for effector CD4 and CD8 T-cells (frequency among CD45RO^+^ T-cells is indicated). (**c**,**d**) Flow cytometric analyses of tumor-infiltrating CD8 T-cells (**c**) or conventional FOXP3-negative CD4 T-cells (**d**) from patient 22 and patient 32, with healthy control (HC)-PBMC shown for comparison. Boxes show frequencies of gated cells, except red boxes that show frequency of T-cells expressing multiple inhibitory receptors (indicated) as defined by Boolean gating.

**Figure 3 cancers-12-03374-f003:**
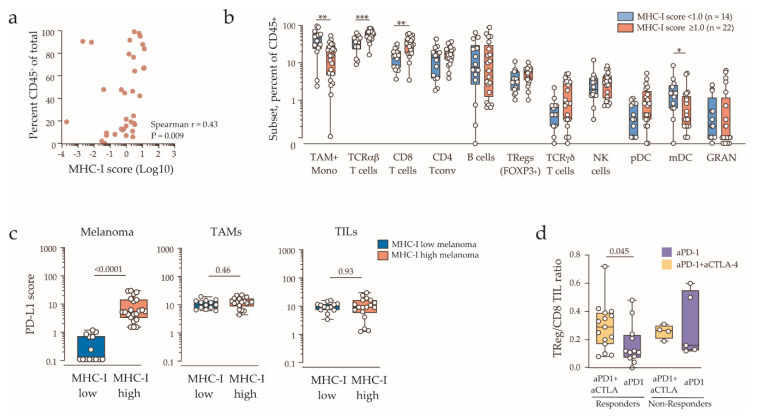
Immune contexture stratified by tumor MHC-I expression. (**a**) Correlation of tumor immune infiltration (CD45^+^ fraction) and melanoma MHC-I expression score, with Spearman’s rank correlation coefficient and the *p* value indicated. (**b**) Immune contexture (percent of CD45^+^ fraction) of MHC-I normal/high (MHC-I score ≥1.0) and MHC-I low/negative tumors (MHC-I score <1.0). Values below 0.1% are shown as zero. Boxes indicate the interquartile range and the median. Statistically significant comparisons (Mann-Whitney’s test) are indicated (* *p* < 0.05; ** *p* < 0.01; *** *p* < 0.001). Tumor-associated macrophages and monocytes (TAM+Mono); T-cells expressing the αβ T-cell receptor (TCRαβ) or the γδ T-cell receptor (TCRγδ); TCRαβ conventional CD4+FOXP3- T-cells (CD4 Tconv) or regulatory CD4+FOXP3+ T-cells (TRegs); natural killer (NK) cells; plasmacytoid dendritic cells (pDC); CD1c+ myeloid DC (mDC); granulocytes (GRAN). (c) PD-L1 expression score on melanoma cells (left panel), tumor-associated macrophages (TAMs, middle panel) and tumor-infiltrating lymphocytes (TILs, right panel) in MHC-I low (blue) and MHC-II high tumors (red). Boxes show the interquartile range and the median, with Mann-Whitney’s p values indicated. (d) Regulatory T-cell/CD8 TIL ratio in tumor biopsies, stratified by therapy and patient’s response. Boxes show the interquartile range and the median, with Mann-Whitney’s *p* value indicated.

**Table 1 cancers-12-03374-t001:** Baseline patient characteristics.

Clinical Characteristics	Single Agent PD-1 (*n* = 17)	Combination Immunotherapy (*n* = 19)
Age		
Median (range)	70 (42–88)	56 (28–72)
Sex, *n* (%)		
Male	8 (47)	14 (74)
Female	9 (53)	5 (26)
Stage (AJCC 8th edition), *n* (%)		
Unresectable IIIc	1 (6)	2 (11)
M1a	3 (18)	0
M1b	3 (18)	2 (11)
M1c	6 (35)	8 (42)
Mutation, *n* (%)		
BRAF V600	2 (12)	9 (47)
BRAF non-V600	0	1 (5)
N/KRAS	7 (41)	2 (11)
BRAF/NRAS wild type	8 (47)	7 (37)
Line of treatment, *n* (%)		
First line	14 (82)	19 (100)
Second line *	3 (18)	0
RECIST response, *n* (%)		
CR/PR	12 (71)	15 (79)
SD/PD	5 (29)	4 (21)
Biopsy site, *n* (%)		
LN	6 (35)	8 (42)
SC	6 (35)	7 (37)
Non-brain visceral	1 (6)	1 (5)
Brain	4 (24)	3 (16)
Pre-treatment biopsy timing		
Median (Range)	85 (0–328)	98 (7–274)
Progression free survival		
Median	NR	NR
6-month	82%	68%
Overall Survival		
Median	NR	NR
12-month	88%	84%

* One patient received combination Dabrafenib and Trametinib in the neoadjuvant/adjuvant setting and two patients received single agent Ipilimumab as first-line therapy. Abbreviations: AJCC, American Joint Committee on Cancer; RECIST, Response Evaluation Criteria In Solid Tumors; CR, complete response; PR, partial response; SD, stable disease; PD, progressive disease; NR, not reached; LN, lymph node; SC, subcutaneous.

**Table 2 cancers-12-03374-t002:** Baseline clinicopathologic characteristics according to tumor MHC-I expression.

Clinical Characteristics	MHC-I Low (*n* = 14)	MHC-I Normal/High (*n* = 22)	*p* Value
Age, *n* (%) Age < 65 Age ≥ 65	10 (71) 4 (29)	10 (45) 12 (55)	0.18
Sex, *n* (%) Male Female	10 (71) 4 (29)	12 (55) 10 (45)	0.48
Biopsy site, *n* (%) SC or LN Non-lung visceral Brain	11 (79) 0 3 (21)	16 (73) 2 (9) 4 (18)	1.0
Mutation, *n* (%) BRAF/N/KRAS mutation BRAF/N/KRAS wild type	9 (64) 5 (36)	12 (55) 11 (45)	0.52
Stage (AJCC 8th edition), *n* (%) IIIc/M1a/b M1c M1d	4 (29) 4 (29) 6 (52)	7 (32) 10 (45) 5 (23)	0.27 *
LDH, *n* (%) Normal >1.0xULN >1.5xULN	6 (43) 8 (57) 5 (36)	17 (77) 5 (23) 1 (5)	0.07 0.02
Years from primary, *n* (%) ≤3 years >3 years Unknown/occult	9 (64) 2 (14) 3 (22)	9 (41) 9 (41) 4 (18)	0.13

MHC-I normal/high, MHC-I score ≥1.0; MHC-I low, MHC-I score <1.0; *p* value, Fisher’s exact test; * M1d versus IIIc/M1a/b/c; Abbreviations: LN, lymph node; SC, subcutaneous; LDH, Lactate dehydrogenase; ULN, upper limit of normal.

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
