# Peer review of "Tumor MHC Expression Guides First-Line Immunotherapy Selection in Melanoma"

_cancers, 2020, doi:10.3390/cancers12113374_

Round 1

Reviewer 1 Report

the study is of major interest and well conducted: bravi

the project should be continued and proposed on larger numbers also retrospectively if possible

Author Response

We thank all reviewers for carefully examining the manuscript and their helpful suggestions.

We agree with the reviewer that our cohort size is small – but as the data are so compelling and potentially valuable we believe that publication as a short report is warranted. We are looking at expanding the data with larger cohorts - but this will take time as it requires banked viable tumor dissociates.

Reviewer 2 Report

This manuscript is well written and experiments and methods are clear. This manuscript further adds to the knowledge base on association of  MHC-I expression on tumor and response to anti-PD-1 monotherapy.

The sample size in the experiment is small but still adequate enough to support the conclusion.

Author Response

(The authors gave the same response as above.)

Reviewer 3 Report

The brief report entitled: “Tumor MHC expression guides first-line immunotherapy selection in melanoma” deals with a hot topic in clinical practice that is the use of immunotherapy to treat patients. The paper is technically sound and well written. Iconography and references are adequate. The manuscript does not have any major flaws. The methods applied are appropriate to the study design. The data are interpreted correctly, and they adequately support the conclusions. The authors present a novel and compelling study that significantly advances the understanding of the effects of immunotherapy treatment on melanoma patients. However, there are a few minor concerns that need to be addressed:

1. How specific is the antibody? The author needs to perform Western blotting on melanoma tissue and benign mole tissue.

2. In addition to anti-PD-1 (Pembrolizumab or Nivolumab) treatment, have these patients received any treatment before? These should be carefully explained.

3. The quality of all pictures needs to be greatly improved. It is currently a bit messy and needs further unification.

4. The manuscript contains some strange English usage and should be further carefully revised.

Author Response

We thank all reviewers for carefully examining the manuscript and their helpful suggestions.

In response to Reviewer 3 comments regarding the specificity of the MHC-class I antibody. We would like to highlight several key points:

  1. The antibody against MHC class I used in this study (clone W6/32) has been used in research since 1987 (Stern, J Immunol 1987, 138:1088; Barnstable Cell 1978, 78:90296) and the specificity verified extensively, including by flow cytometry and WB assays.
  2. We have provided evidence of the antibody specificity in Figure 1c, d. In particular, we show that patient-derived melanoma cells deficient for the invariant component of the MHC class I complex, beta-2-microglobulin (as confirmed by WB using a different antibody), lack cell surface expression of MHC class I in a flow cytometric assay with W6/32 antibody (Fig.1 c,d). We state on p. 5, lines 139-144: “The validity of this assay was confirmed with the identification of MHC-I loss (MHC-I score <0.001, Figure 1c) in independent melanoma biopsies taken from two patients who progressed while on anti-PD-1 treatment. Loss of MHC-I expression in both tumors was associated with loss of the MHC-I invariant chain B2M in the corresponding biopsy-derived melanoma cell lines (Figure 1d)”.

With regards to [patients receiving] prior systemic therapy, we have added the following information (changes highlighted):

p.3 lines 101-104: Prior treatment was administered in 3/17 (18%) patients who received anti-PD-1 alone (ipilimumab in two patients and combination BRAF/MEK inhibitor in one patient), whereas no patients who received combination immunotherapy had prior systemic therapy

Table 1, Line of treatment: Changed from “≥ second line”, to “second line”, with an explanatory footnote “One patient received combination dabrafenib and trametinib in the neoadjuvant/adjuvant setting and two patients received single agent ipilimumab as first-line therapy”.

We have reviewed the manuscript and made minor editorial revisions throughout to improve readability.

Reviewer 4 Report

In this brief report, the Authors presents results that emphasize the importance of MHC-I expression for patient response to anti-PD-1 monotherapy, and additionally they provide a rationale for the selection of combination immunotherapy as the first-line treatment in MHC-I low melanoma. The Authors use clear experimental model and demonstrate logical workflow for determination of the levels of markers with potential to be used as a strategy for patient selection for immunotherapy. This report is well written, data are very clearly presented, and conclusions are sufficiently supported by the results. Please take care of Supplemental Material as it contains unnecessary parts eg., references or excessive words.

Author Response

We thank all reviewers for carefully examining the manuscript and their helpful suggestions.

We have added a missing reference to the Supplementary data, removed the subtitles and modified the legend to Figure S2 (f).

Round 2

Reviewer 3 Report

Accept in present form